# Descriptive Statistics and Genome-Wide Copy Number Analysis of Milk Production Traits of Jiangsu Chinese Holstein Cows

**DOI:** 10.3390/ani14010017

**Published:** 2023-12-19

**Authors:** Hao Zhu, Xubin Lu, Hui Jiang, Zhangping Yang, Tianle Xu

**Affiliations:** 1Joint International Research Laboratory of Agriculture and Agri-Product Safety, Yangzhou University, Yangzhou 225009, China; mx120210848@stu.yzu.edu.cn (H.Z.); yzp@yzu.edu.cn (Z.Y.); 2College of Animal Science and Technology, Yangzhou University, Yangzhou 225009, China; lxb@yzu.edu.cn; 3Center for Quantitative Genetics and Genomics, Aarhus University, 8000 Aarhus C, Denmark; dx120210160@stu.yzu.edu.cn; 4International Joint Research Laboratory, Universities of Jiangsu Province of China for Domestic Animal Germplasm Resources and Genetic Improvement, Yangzhou 225009, China

**Keywords:** milk production traits, genetic correlation, phenotypic correlation, copy number variation analysis

## Abstract

**Simple Summary:**

The aim of this study was to describe the basic status of milk production traits of Chinese Holstein cows in Jiangsu, and locate the QTLs and functional genes that affect the five milk production traits (milk yield, milk fat percentage, milk fat yield, milk protein percentage, and milk protein yield) of first-born cows. The five milk production traits showed significant phenotypic positive correlation and high genetic positive correlation among the three parities, and we identified 1731 CNVs and 236 CNVRs in the 29 autosomes of 4173 Holstein dairy cows, and 19 CNVRS were significantly associated with milk production traits. The study introduced the basic status of the milk production traits of Chinese Holstein cows in Jiangsu province, and 19 CNVRs were identified to be significantly related to milk production traits of Chinese Holstein cows in the first fetus. This study will provide a theoretical basis for molecular marker-assisted selection of dairy cows, help to analyze the genetic structure of milk production traits of dairy cows, and contribute to the genetic improvement of Chinese Holstein cattle in Jiangsu.

**Abstract:**

Milk production traits are the most important quantitative economic traits in dairy cow production; improving the yield and quality of milk is an important way to ensure the production efficiency of the dairy industry. This study carried out a series of in-depth statistical genetics studies and molecular analyses on the Chinese Holstein cows in the Jiangsu Province, such as descriptive statistics and copy number variation analysis. A genetic correlation, phenotypic correlation, and descriptive statistical analysis of five milk production traits (milk yield, milk fat percentage, milk fat yield, milk protein percentage, and milk protein yield) of the dairy cows were analyzed using the SPSS and DMU software. Through quality control, 4173 cows and their genomes were used for genomic study. Then, SNPs were detected using DNA chips, and a copy number variation (CNV) analysis was carried out to locate the quantitative trait loci (QTL) of the milk production traits by Perl program software Penn CNV and hidden Markov model (HMM). The phenotypic means of the milk yield, milk fat percentage, milk fat mass, milk protein percentage, and milk protein mass at the first trimester were lower than those at the other trimesters by 8.821%, 1.031%, 0.930%, 0.003%, and 0.826%, respectively. The five milk production traits showed a significant phenotypic positive correlation (*p* < 0.01) and a high genetic positive correlation among the three parities. Based on the GGPBovine 100 K SNP data, QTL-detecting research on the fist-parity milk performance of dairy cows was carried out via the CNV. We identified 1731 CNVs and 236 CNVRs in the 29 autosomes of 984 Holstein dairy cows, and 19 CNVRs were significantly associated with the milk production traits (*p* < 0.05). These CNVRs were analyzed via a bioinformatics analysis; a total of 13 gene ontology (GO) terms and 20 Kyoto Encyclopedia of Genes and Genomes (KEGG) pathways were significantly enriched (*p* < 0.05), and these terms and pathways are mainly related to lipid metabolism, amino acid metabolism, and cellular catabolic processes. This study provided a theoretical basis for the molecular-marker-assisted selection of dairy cows by developing descriptive statistics on the milk production traits of dairy cows and by locating the QTL and functional genes that affect the milk production traits of first-born dairy cows. The results describe the basic status of the milk production traits of the Chinese Holstein cows in Jiangsu and locate the QTL and functional genes that affect the milk production traits of the first-born cows, providing a theoretical basis for the molecular-marker-assisted selection of dairy cows.

## 1. Introduction

Milk production traits are the most important economic traits of dairy cows, including milk yield, milk fat percentage, milk fat amount, milk protein percentage, and milk protein amount [1]. Milk production is a quantitative trait, and its performance is influenced by heredity and the environment. The genetic effect can be divided into the additive effect, epistatic effect, and dominant effect [2]. The additive effect is the accumulation of the genotype values of multiple minor genes that affect the quantitative traits, which can be truly inherited by offspring. Therefore, it is also called the breeding value of traits, and it is the main component of the phenotypic value of traits [3]. The main purpose of dairy breeding is to breed the individuals with the best genetic performance (the highest breeding value) through the selection of the economic traits, health traits, and reproductive traits of dairy cows so that their offspring, as a whole, contain more desired genes or contain a higher probability of desired genotypes than the current selection. Thus, the purpose of improving the genetic advantage of individual traits and the population production level can be achieved [4]. In early dairy breeding work, people mainly relied on the methods of Dairy Herd Improvement (DHI) and progeny determination to evaluate the production performance and genetic improvement of the dairy herd. Although effective, there are also problems, such as long generation intervals, high costs, and slow genetic progress, in the face of difficult-to-measure traits [5]. With the development of the quantitative genetics theory and practice as well as the rapid progress of computer technology and biotechnology, dairy workers have made genetic improvements to the economic traits of dairy cows based on traditional genetics combined with the theories and achievements of molecular genetics and bioinformatics [6].

Since it was recognized that single-nucleotide polymorphisms (SNPs) have the function of regulating the expression of single or multiple genes for a variety of important economic traits in dairy cows, cow genetics have made great progress [7,8]. However, recent studies have shown that genomic variations are caused by SNPs, with a significant impact on the performance of various traits of dairy cows, such as their milk yield, reproduction, health, and feed efficiency [9,10]. Structural variation includes translocation, inversion, copy number variation (CNV), etc. [11]. A CNV is the most common type of structural variation in human, plant, and animal genomes, and it refers to an unbalanced structural variation in the genome of an individual base in the form of DNA fragment loss or acquisition [12]. The total number of nucleotides covered by SNVs greatly exceeds the total number of single nucleotide polymorphisms, which greatly enriches the diversity of genomic genetic variation [13]. CNVs can lead to a phenotypic variation related to single-gene diseases, complex diseases, or quantitative traits via molecular mechanisms such as a gene structure change, gene fusion, and the dose effect [14]. For example, the genome-wide association study (GWAS), based on CNVs, identified the functional genes associated with feed conversion and growth and development [15]. Therefore, CNVs are one of the important factors that cause genetic and phenotypic variations as well as an important genetic resource for biological evolution [16]. The presence of CNVs in the protein-coding region of the genome may change the protein function, while the presence of CNVs in the regulatory region of the genome may change the expression level of the genes [17].

Previous CNV detection in most populations was based on comparative genomic hybridization (GCH) and SNP genotyping microarrays [18]. With the development and cost reduction in next-generation sequencing (NGS) technology, CNV detection and analysis can be performed using whole-genome sequencing, which has a higher CNV detection rate than genotyping because it covers the whole genome [19]. At present, there are many algorithms and pieces of software that can be used to identify genomic CNVs and CNVRs [20,21]. The PennCNV software uses the hidden Markov model (HMM) algorithm to identify CNVs; it can integrate the signal strength value (the Log R ratio (LRR)), the B allele frequency (BAF), the population frequency of the B allele (PFB) of SNPs, the distance between adjacent SNPs, the SNP allele frequency, pedigree information, etc. At the same time, the software can fit the genomic GC content of the regression model to overcome the interference of the genomic wave during CNV recognition, improving the accuracy of CNV recognition [22].

It has been found that CNVs evolve about 2.5 times faster than SNPs, indicating that CNVs can cause organisms to better adapt to different environments [23]. At present, there have been many research reports on bovine CNVs. Silva confirmed that the incidence of the CNV overlap in common cattle from Europe and the Middle East is higher than that in Zebu cattle from the Indian subcontinent and speculated that Zebu cattle have a greater diversity in CNVs [24]. Pezar believed that the variation in the CNV quantity may be related to differences in the effective population size and selection process among different populations of cattle [25]. Bae used the PennCNV software to identify a total of 368 CNVRs from 265 Korean Hanwoo cattle based on a 50 K SNP chip covering a 15.8 Mb genome without considering the lineage information or genome waves [26]. At the same time, it has been reported that CNVs are closely related to the milk production traits of Holstein cattle [27].

In this study, based on the population data of the Chinese Holstein cattle in the Jiangsu area, we explored the lactation performance and (covariance) components of the milk production traits in different parities of the cattle and used CNVs to detect the QTL of the milk production traits of the first parity of the Chinese Holstein cattle in Jiangsu.

## 2. Materials and Methods

### 2.1. Ethical Statement

The procedures of hair sample and phenotypic trait data collection were carried out according to the plan proposed by the Ministry of Agriculture of the People’s Republic of China and the Chinese Animal Protection Commission. The study was approved by the Ethics Committee for Animal Researchers of Yangzhou University (License No. SYXK(SU)IACUC 2012-0029). No animals were harmed during the study.

### 2.2. Phenotypic Data Collection and Quality Control

We collected population genetic improvement (DHI) records from 15,126 lactating cows in 4 medium and large farms in Jiangsu, requiring each cow to have at least 7 DHI records during the complete lactation period. Then, a total of 149,065 DHI records were collected. In the DHI records, five traits (milk yield (MY), milk fat percentage (MFR), milk fat yield (MFR), milk protein percentage (MPR), and milk protein yield (MPY)) were selected for genetic assessment in this study. Before analysis, all DHI records were quality-controlled as follows: (1) only 1–3 parity data were retained; (2) the first calving month is between 22 and 36 months; (3) only records of lactation days (DIM) from 5 to 305 days were kept; (4) the daily milk production was maintained in the range of 5–80 kg, and the somatic cell score (SCS) was maintained in the range of 0–9; (5) for each trait, values that exceeded the mean plus or minus a range of three standard deviations were removed. Before analysis, all 5 milk production traits after quality control were calibrated to 305-day performance based on the methods we have [28]. The requirement for convergence of the model used is that the norm of the update vector is less than 1.0 × 10^−7^ or the norm of the gradient vector (AI) is less than 1.0 × 10^−6^. The kinship matrices of animals need to be constructed based on pedigrees [29].

### 2.3. Analysis of Phenotypic and Genotypic Parameters

The paired Pearson coefficient and descriptive statistics of the phenotypic dataset were analyzed using the computer software SPSS (v16.0, Chicago, IL, USA). The variance component estimates were calculated using the DMU software (V5.6, Aarhus, Denmark). We estimated genetic parameters in the following multiple-trait animal model:y=1μ+Za+e
where *y* is the 305-day performance; *Z* is the random effects structure matrix; *a* is the random effect vector; *e* is a random residual vector [30]. Putting the resulting variance components into the following formula to calculate genetic and phenotypic correlations:$$r_{A}=\frac{Cov(\alpha_{1},\alpha_{2})}{\sqrt{\sigma_{\alpha_{1}}^{2}\sigma_{\alpha_{2}}^{2}}}\;r_{p}=\frac{Cov(P_{1},P_{2})}{\sqrt{\sigma_{P_{1}}^{2}\sigma_{P_{2}}^{2}}}$$
$r_{A}$ was the genetic correlation among adjusted traits; $r_{p}$ was a phenotypic correlation among adjusted traits; $Cov(\alpha_{1},\alpha_{2})$ is the additive effect covariance of traits α_1_ and α_2_; $\sigma_{\alpha_{1}}^{2}$ is the additive variance of the trait $\alpha_{1}$; $\sigma_{\alpha_{2}}^{2}$ is the additive variance of the trait $\alpha_{2}$; $Cov\left(P_{1},P_{2}\right)$ is the phenotypic covariance of $P_{1}$ and $P_{2}$; $\sigma_{P_{1}}^{2}$ is the phenotypic variance of the trait $P_{1}$; $\sigma_{P_{2}}^{2}$ is the phenotypic variance of the trait $P_{2}$.

### 2.4. Genotyping Data and Quality Control

Hair follicle samples were collected from the tail roots of selected cows with at least 50 hair follicles per cow. The sample was sent to the company and genotyped using the GGPBovine 100 K SNP Chips (Newqin Biotechnology, Shanghai, China), in which the bovine ARS-UCD1.2 assembly genome was used as the reference genome. Then, PLINK software was used for genotype quality control according to the following requirements: (1) SNPs with a detection rate >90% were retained; (2) individuals with an SNP detection rate greater than 95% were retained; (3) sites with minimum allele frequency (MAF) < 0.05 were proposed; (4) SNPs that deviated significantly from Hardy–Weinberg equilibrium (HWE) were removed (*p* < 1.0 × 10^6^) [31].

### 2.5. Population Structure Analysis

Since the subgroups in this study came from four different pastures in Jiangsu, in order to identify the population structure of the subgroups, we used the SNPRelate package in R (v 4.0.2, Auckland, New Zealand) to conduct cluster analysis and identity by state (IBS) analysis for the populations, and to determine the effect of population stratification on phenotypic observations.

### 2.6. Detection of CNVR

In this study, the Perl program software Penn CNV and the hidden Markov model (HMM) were used to infer the CNV of the Chinese Holstein cattle population in Jiangsu. During the process of inference, pennCNV software integrates SNP signal strength file information, SNP location information, population pedigree information, etc. And the GC content of each SNP within the 1 Mb range of the genome can be substituted to correct the effect of genomic waves on CNVs inference caused by different genomic G/C content. The signal strength (Log R Ratio, LRR) and allele strength ratio (B allele frequency, BAF) of all SNPs were generated by Illumina BeadStudio3.5 software (Illumina, San Diego, CA, USA). Genotypic populations have complete genealogical records. The PFB (B allele of population frequency) file is generated based on the BAF of each marker in this population. The signal strength of each SNP affected by genomic waves is adjusted according to the GC content of the 500 kb genomic region on either side of it [32].

In order to improve the accuracy of CNV results detection, we used Perl programming language to conduct quality control on the obtained individual copy number variation results. The quality control criteria were as follows: (1) LRR standard deviation was less than 0.3; (2) the standard deviation of BAF was less than 0.01; (3) the absolute value of Waviness factor (GCWF) was less than 0.05; (4) confidence score was greater than 10; (5) the number of SNPs in each detected CNV were greater than or equal to 10; (6) it had to be detected simultaneously in at least two individuals [33].

All CNVs after quality control were merged according to the following criteria: if adjacent CNVs overlap and the overlap area is not less than 20% of the total length, and these CNVs had the same copy number variation state (increased or missing), they were merged into a new CNV. If at least one SNP overlapped between adjacent CNVs, the CNVs obtained by quality control were merged into the CNV Region (CNVR). Since the starting and ending position information of each CNVR is not fixed, we took the starting site of the CNV at the front of the chromosome as the starting site of the merged CNVR, and the ending site of the CNV at the back of the chromosome as the ending site of the merged CNVR. CNVRs were used for subsequent analysis of association with milk production traits.

### 2.7. Association Analysis Based on CNVR

The general linear model (GLM) of CNVR Ruler software was used to conduct a correlation analysis of the milk production characteristics (milk yield, milk fat percentage, milk fat amount, milk protein percentage, and milk protein amount) obtained in the CNVR region, respectively. Five milk production traits were calibrated to 305-day performance based on the models we have [28]. The GLM model used is as follows:*y* = *μ* + *G*_i_ + *e*_i_

Among them, *y* was the corrected phenotypic values of milk yield, milk fat percentage, milk fat yield, milk protein percentage, and milk protein yield at 305 days of the first gestational term. *μ* is the population mean; *G*_i_ is the genotype effect of the ith CNVR; *e*_i_ is the random residual effect of the ith CNVR, and the result is corrected using the FDR (q-value) method.

### 2.8. Bioinformatics Analysis of Candidate Genes

Annotation and gene function analysis were performed using the online Database for Annotation, Visualization, and Integrated Discovery (DAVID) software (https://david.ncifcrf.gov/tools.jsp), accessed on 25 June 2022.

We performed kyoko Encyclopedia of Genes and Genomes (KEGG) pathway analysis and Gene ontology (Gene Ontology, GO) analysis of CNVRs annotated in exonic regions by KOBA3.0 annotation outside show (http://kobas.cbi.pku.edu.cn/), accessed on 25 June 2022 the results only retained the path entries with corrected *p*-values less than 0.05 were retained.

## 3. Results

### 3.1. Descriptive Statistics of Milk Production Traits of the Chinese Holstein Cattle in Jiangsu

Through quality control, 94,595 measurement day records for 9828 cows out of 15,126 lactating cows were retained for subsequent analysis. The descriptive statistics of the milk production traits of the Chinese Holstein cattle in Jiangsu are shown in Table 1. Among them, for the milk yield trait, the number of measurement day records for first-born cows accounted for 49.02% of the total number of daily records, and the number of first-born cows accounted for 65.46% of the total. For the milk fat percentage, milk fat content, milk protein percentage, and milk protein content, the number of measurement day records for first-born cows accounted for 49.39% of the total number of daily records, and the number of first-born cows accounted for 64.02% of the total. The phenotypic mean of the five milk production traits of the first parity was lower than those of the other parities. The standard errors of the five milk production traits in the first trimester were lower than those in the other trimesters. In addition to the milk protein percentage, the skewness of the milk yield, milk fat percentage, milk fat content, and milk protein content was closest to zero in the first parity.

### 3.2. Phenotypic and Genetic Correlation among the Different Parities’ Milk Production Traits of the Dairy Cows in Jiangsu

Table 2 shows the phenotypic and genetic correlation of the milk production traits among the different parities of the Chinese Holstein cattle in Jiangsu. The five milk production traits studied in this paper showed a significant phenotypic positive correlation (*p* < 0.01) and a high genetic positive correlation among the three parities. In addition, the phenotypic correlation and genetic correlation of most of these traits between the first and third parity were higher than those between the first and second parities (the phenotypic correlation between the first and third parity was lower than that between the first and second parities). We also found that the phenotypic correlation values of the five milk production traits were smaller than the corresponding genetic correlation values between the different parities.

### 3.3. Group Structure Analysis

Through genotype quality control, 4173 individual cows and 87,598 SNPs were retained. A cluster analysis and state homology (IBS) were performed on 4173 dairy herds using chip data, and neither method significantly distinguished the study groups, indicating a high correlation of individuals within the study groups (Figure 1A,B).

### 3.4. Genome Copy Number Variation in the Chinese Holstein Cattle in Jiangsu

In this study, bovine ARS-UCD1.2 was used as the reference genome, and a total of 1731 CNVs were detected in the bovine autosomes after quality control via the PennCNV software. Among them, there were 14 CNVs with a copy number of zero (about 0.80%), and the average length was 207,115 bp. There were 68 CNVs with a copy number of one (about 3.93%), and the average length was 302,899 bp. There were 1550 CNVs with a copy number of three (89.54%), and the average length was 440,638 bp. There were 99 CNVs with a copy number of four (about 5.73%) and an average length of 422,555 bp. Copy number replication (types 3 and 4) accounts for more than 95% of the copy number variations (Table 3). Of all the 1731 CNVs detected, CNVs between 200 and 500 kb in length accounted for the majority (Figure 2A), and the average length of the CNVs detected in the duplication type was larger than that of the deletion type (Figure 2B); meanwhile, there were more copy number variations on chromosomes 18 and 21 (Figure 2C,D).

### 3.5. Regional Distribution of Copy Number Variations in the Chinese Holstein Cattle in Jiangsu

In this study, the PennCNV software was used to analyze the copy number variation in the 100 K chip data of 4173 Holstein cattle in Jiangsu and 1731 CNVs were obtained. After the combination, 236 copy number variation regions (CNVs) were obtained after the CNVRs expressed only in a single sample were removed. Duplicate CNVRs occupy the vast majority of all CNVRs. The proportion of copy number variations in the 29 autosomes of the dairy cows ranged from 3.9% to 20.7% among which chromosome 28 (20.7%) had the largest copy number variation region, while chromosome 8 (3.9%) had the smallest copy number variation region. There were six chromosomes with copy number variation regions covering more than 10% (Figure 3).

### 3.6. Association Analysis of the CNVRs and Milk Production Traits of the Chinese Holstein Cattle in Jiangsu

After FDR correction, a total of 19 CNVRs was significantly correlated with five milk production traits (milk yield, milk fat percentage, milk fat volume, milk protein percentage, and milk protein volume) in the first parity of the dairy cows in Jiangsu. SNVRs significantly associated with milk production traits were mainly concentrated on chromosomes 11, 14, and 17. The study did not find any CNVRs significantly associated with milk protein volume.

Three CNVRs, located on chromosomes 17, 17, and 11, were significantly correlated with the milk yield, and their copy number variation types were “deletion”, “deletion”, and “mixed”. Four CNVRs were significantly correlated with the milk fat percentage, and they were located on chromosomes 14, 15, 21, and 3, respectively. Their copy number variation types were “mixed”, “duplicate”, “deletion”, and “duplicate”. Six CNVRs were significantly correlated with the milk fat content, and they were located on chromosomes 17, 11, 12, 14, 17, and 18, respectively. Their copy number variation types were “duplicate”, “mixed”, “duplicate”, “mixed”, “deletion”, and “duplicate”. Six CNVRs, located on chromosomes 11, 17, 17, 11, 5, and 5, were significantly correlated with the milk protein rate, and their copy number variation types were “mixed”, “deletion”, “deletion”, “mixed”, “mixed”, and “mixed” (Table 4). The study did not find any CNVRs significantly associated with milk protein volume.

### 3.7. Enrichment Analysis of Candidate Genes

In order to better understand the biological function of the 19 CNVRs, we performed a KEGG pathway analysis and a GO biological function enrichment analysis for the genes within these CNVR regions and upstream and downstream these regions within 200 Kb, and a total of 216 genes were selected (Appendix A). These genes were significantly enriched into 20 KEGG pathways; among them, AKT1, VAV2, and NOTCH1 had more repeats (Table 5). After removing the GO terms concentrated in only one gene, a total of 13 GO terms were selected, which are mainly involved in biological processes such as amino acid metabolism, cellular catabolism, and lipid metabolism, and the genes involved are DGAT1, GPT, SCO2, DBH, etc. (Figure 4; Appendix A).

## 4. Discussion

In this study, the variance components, and the genetic and phenotypic correlations of the five milk production traits (the milk yield, milk fat percentage, milk fat yield, milk protein percentage, and milk protein yield) among the different parities of the Chinese Holstein cattle in Jiangsu were evaluated. The phenotypic and genetic correlation between the milk production traits of the Chinese Holstein cattle in Jiangsu was similar to some domestic and foreign research results [34,35]. The same milk production trait showed a significant positive phenotypic correlation and a high positive genetic correlation at one to three parities. Therefore, it is feasible to evaluate the overall performance of dairy cows according to the performance of milk production traits at one parity.

Copy number variations are one of the common forms of genomic structural variations in plants and animals, and CNV variations cover a large number of SNPs; therefore, they often cause changes in the gene sequences of large segments of biological genomes, resulting in changes in the gene structure, changes in the expression dose, and even a loss of gene function, thus affecting the phenotype [36]. Currently, there are PennCNV, cnvPartition, and QuantiSNP as well as other software used to infer CNVs and CNVRs based on SNP chip data [37,38]. Studies have shown that the PennCNV software can use more information (such as the relatives, the total signal intensity of the chip data, the B allele frequency, etc.) in the process of CNV inference, and its algorithm is more reliable, resulting in less false positives [39]. In this study, the PennCNV software was used to detect a total of 1731 CNVs and 236 CNVRs in 29 autosomes of the Chinese Holstein cattle in Jiangsu among which the duplication types of the CNVs and CNVRs were higher than the deletion types (Figure 3). This may be related to the selection and breeding of the cows [40].

CNVs can play a role in variety formation and environmental adaptation, and copy number differences between different varieties and the same variety in different environments can lead to quantitative trait variations [41,42]. According to some reports, the size and quantity of the CNVRs of the Chinese Holstein cattle in different regions also vary greatly. Jiang identified 367 CNVRs in the Chinese Holstein cattle through a PennCNV analysis of 50 K SNP chip genotyping data from 96 Chinese Holstein cattle in Beijing [43]. In this study, 236 CNVR regions of the Chinese Holstein cattle in Jiangsu were excavated using 100 K chips. Such differences in the number of CNVRs may have been caused by differences in the environment, chip density, population size, and structure [44]. Although SNP chips from cattle can be used for CNV detection, the SNP probes on the chips are neither dense enough nor evenly distributed enough to obtain an accurate high-resolution map of the copy number variations in the bovine genome. The average interval between adjacent SNPs in the bovine 100 K SNP chip was about 25 Kb, and the main application scenarios of this chip are genome-wide association analysis and genome selection; additionally, a large part of the probes may be located outside the CNVRs [45]. Therefore, only CNVRs with a certain length may be found in this study, and all CNVRs with a short length and no probe label are difficult to detect based on chip data alone. Studies on humans have shown that smaller-length CNVRs appear more frequently in the genome than larger-length CNVRs [46,47]. With the application of bovine high-density 800 K chips or next-generation sequencing methods, it is expected that more and more cross-genome CNVs and CNVRs will be identified.

It has been confirmed that the CNVs in the dairy cow genome are closely related to their milk production traits. For example, Gao found that the functional differences of the CNVs and CNVRs in the genomes of cows with a high milk fat percentage and high milk protein percentage and those of cows with a low milk fat percentage and low milk protein percentage mainly involved fat and protein metabolism and other related biological processes. Based on this, 10 candidate genes related to the milk production traits of dairy cows were identified for the subsequent molecular breeding of the dairy cows [48]. This study also identified 19 CNVRs that were significantly associated with the milk production traits of the Chinese Holstein cattle, and these CNVR regions could be used as the key genomic candidate regions for the milk production traits of the dairy cows.

Based on the existing data, this study only preliminarily explored the copy number variation areas and individual production traits. A total of 19 copy number variation regions were found to have significant effects on five milk production traits of Jiangsu Holstein dairy cows. The study did not find any CNVRs significantly associated with milk protein volume, which requires further exploration.

A KEGG pathway analysis was performed on 216 genes in the 19 screened CNVRs that were closely related to the first fetal milk production traits of the dairy cows. In order to find the core biological categories and corresponding genes that were closely related to the milk production traits, we eliminated the genes that were enriched in only one biological category. Finally, 13 GO terms with significantly enriched biological functions were retained, which mainly involved lipid synthesis, amino acid metabolism, glucose metabolism, etc., and were mostly related to the milk production performance of the dairy cows [49,50]. There were also 10 candidate genes that were highly involved in the enrichment of these GO terms. The DGAT1 gene is a basic metabolic enzyme that plays an important role in the biosynthesis of triglyceride, the metabolism of triglyceride, and the digestion and absorption of the fat in milk. Many studies have revealed the relationship between DGAT1 and the milk production performance of dairy cows. In particular, the polymorphism of K232A in DGAT1 can significantly affect the expression of the milk fat and milk protein content of dairy cows [51,52]. *VPS28* may regulate milk fat synthesis through ubiquitination in bovine mammary epithelial cells [53]. *MAF1* has been identified as a candidate gene associated with the milk yield and milk protein content in Canadian Holstein cattle studies [54]. By using mid-infrared mass spectrometry and GWAS techniques, *GPT* was listed as a candidate gene for the characteristics associated with the production of French Holstein milk cheese [55]. At the same time, we found that *DAGT1* and *GPT* reappeared in the GO enrichment results, and they may be important candidate genes affecting the performance of the first litter of the Chinese Holstein cattle in Jiangsu.

## 5. Conclusions

This study introduced the basic status of the milk production traits of the Chinese Holstein cows in the Jiangsu Province, and 19 CNVRs were identified to be significantly related to the milk production traits of the Chinese Holstein cows in the first parity. SNVRs significantly associated with milk production traits were mainly concentrated on chromosomes 11, 14, and 17. CNVR_156 on chromosome 17 was found to be significantly correlated with milk yield, milk quality, and milk protein percentage. CNVR_124 on chromosome 14 was significantly associated with milk fat. CNVRs significantly correlated with milk fat percentage and milk fat content did not show overlapping regions.

We also found that *AKT1*, *VAV2*, *NOTCH1*, *DGAT1*, *GPT*, *SCO2*, and *DBH* may be important candidate genes affecting the performance of the first litter of the Chinese Holstein cattle in Jiangsu. This study provides a theoretical basis for the molecular-marker-assisted selection of dairy cows, helps analyze the genetic structure of the milk production traits of dairy cows, and contributes to the development of the quantitative genetics of dairy cows and the genetic improvement in the Chinese Holstein cattle in Jiangsu.

## Figures and Tables

**Figure 1 animals-14-00017-f001:**
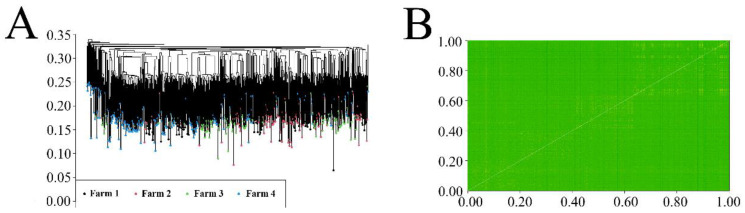
Population cluster analysis and identity-country analysis. (**A**) Cluster analysis; (**B**) identity-country analysis.

**Figure 2 animals-14-00017-f002:**
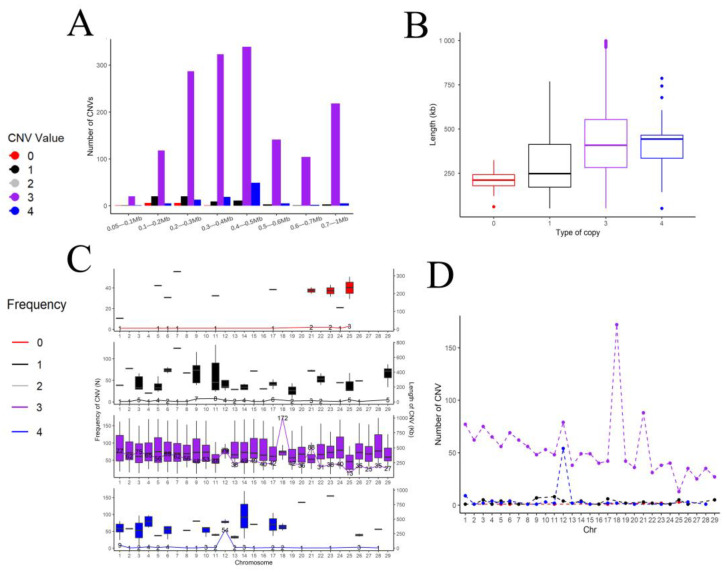
Copy number variation distribution in the bovine reference genome ARS-UMD1.2. (**A**) Length distribution of CNVs; (**B**) size distribution of different types of CNVs; (**C**,**D**) distribution of different types of CNVs on autosomes of bovine.

**Figure 3 animals-14-00017-f003:**
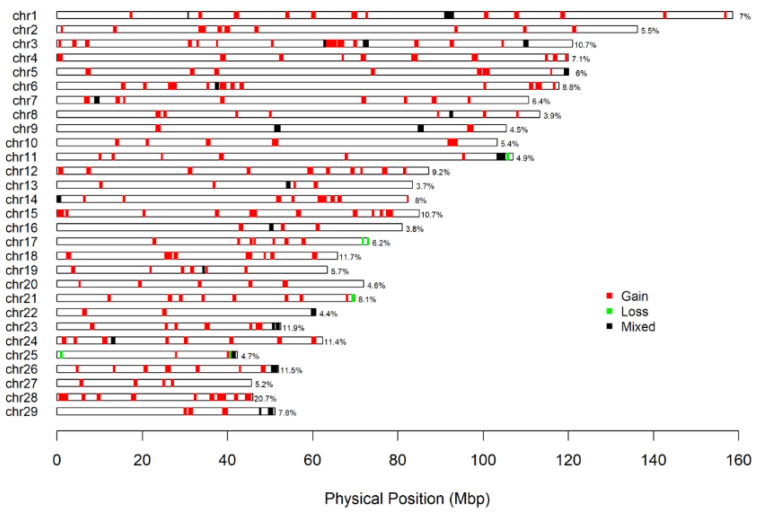
Genome distribution of Holestin CNVRs in Jiangsu province. Red = duplicate; green = deletion type; black silk = mixed type.

**Figure 4 animals-14-00017-f004:**
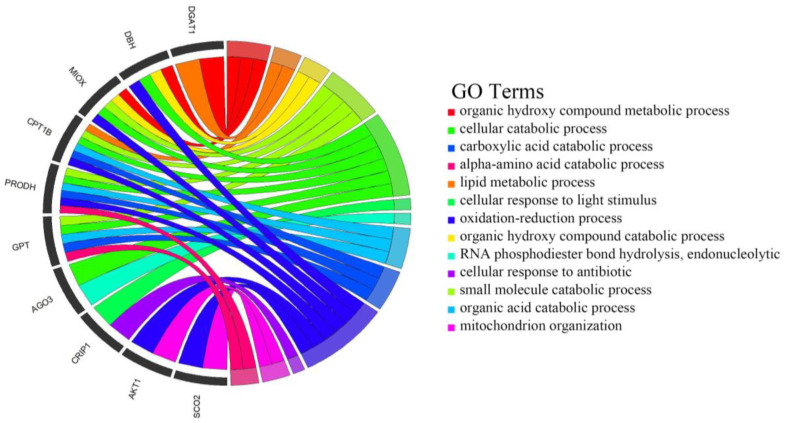
Significant Gene Ontology (GO) terms enriched by the genes within CNVRS. After quality control, 13 GO terms that are enriched by 10 genes are revealed in the circus plots (*p* < 0.05).

**Table 1 animals-14-00017-t001:** Descriptive statistics of milk production traits in Chinese Holstein cows in parity 1–3 in Jiangsu.

Traits	Parity	NI	NR	Mean	SD	Median	Min	Max	Skew	Kurtosis
MY (kg)	1	6437	46,749	30.39	7.98	30.70	5.00	75.00	−0.08	0.25
	2	3402	24,864	36.28	10.79	36.00	5.00	80.00	0.12	0.17
	3	3124	23,162	36.94	10.95	36.50	5.00	79.00	0.10	−0.10
	1–3	9834	95,375	33.33	6.23	33.00	5.00	80.00	0.31	0.43
MFR (%)	1	6263	45,814	3.84	0.96	3.79	0.33	9.99	0.48	1.55
	2	3196	23,785	3.91	1.02	3.87	0.32	9.99	0.50	1.34
	3	3088	23,104	3.97	1.04	3.92	0.36	9.57	0.55	1.23
	1–3	9497	92,763	3.88	1.00	3.82	0.30	9.99	0.52	1.46
MFY (kg)	1	6263	45,814	1.17	0.40	1.14	0.06	4.58	0.70	1.62
	2	3196	23,785	1.41	0.53	1.35	0.08	4.90	0.77	1.41
	3	3088	23,104	1.46	0.55	1.40	0.08	5.74	0.77	1.47
	1–3	9497	92,763	1.29	0.50	1.23	0.60	5.74	0.93	2.00
MPR (%)	1	6263	45,814	3.27	0.36	3.25	0.30	9.46	0.74	6.50
	2	3196	23,785	3.29	0.38	3.26	0.75	7.27	0.71	3.43
	3	3088	23,104	3.30	0.39	3.27	1.22	7.22	0.62	2.94
	1–3	9497	92,763	3.28	0.37	3.25	0.30	9.46	0.75	4.95
MPY (kg)	1	6263	45,814	1.00	0.25	1.00	0.11	2.43	−0.01	0.59
	2	3196	23,785	1.18	0.34	1.18	0.17	3.27	0.22	0.83
	3	3088	23,104	1.21	0.33	1.2	0.13	2.76	0.07	0.26
	1–3	9497	92,763	1.09	0.32	1.07	0.10	4.58	0.36	1.02

Parity = parity; NI = number of recorded individuals; NR = number of test-day records; Mean = mean; SD = standard deviation; Median = median; Min = minimum; Max = maximum; Skew = skewness; Kurtosis = Kurtosis.

**Table 2 animals-14-00017-t002:** Phenotypic and genetic correlations of a trait in different parities of Chinese Holstein dairy cows in Jiangsu Province.

Traits	Parity	Parity 1	Parity 2	Parity 3
MY	1	1.000	0.478 **	0.512 **
	2	0.596 (0.098)	1.000	0.453 **
	3	0.857 (0.088)	0.867 (0.102)	1.000
MFR	1	1.000	0.380 **	0.421 **
	2	0.913 (0.078)	1.000	0.419 **
	3	0.963 (0.141)	0.989 (0.143)	1.000
MFY	1	1.000	0.362 **	0.358 **
	2	0.623 (0.158)	1.000	0.324 **
	3	0.779 (0.171)	0.975 (0.205)	1.000
MPR	1	1.000	0.444 **	0.554 **
	2	0.902 (0.054)	1.000	0.546 **
	3	0.963 (0.053)	0.985 (0.078)	1.000
MPY	1	1.000	0.420 **	0.436 **
	2	0.552 (0.117)	1.000	0.399 **
	3	0.858 (0.122)	0.815 (0.144)	1.000

The data in the table is expressed as R-square. Data below the diagonal lines are genetic correlations, data above the diagonal lines are phenotypic correlations, and the diagonal lines are phenotypic correlations and genetic correlations; ** means *p* < 0.01.

**Table 3 animals-14-00017-t003:** Statistical information of the CNVs in the bovine reference genome ARS-UMD1.2.

CNV Type	Counts	Length	Min-Length	Max-Length
0	14	207,115	60,725	323,251
1	68	302,899	50,880	768,584
3	1550	440,638	50,683	998,290
4	99	414,047	50,735	980,071

0 and 1 represent deletions; 3 and 4 represent duplications.

**Table 4 animals-14-00017-t004:** Information of CNVRs that significantly associated with milk-related traits in Holstein dairy cows in Jiangsu.

Traits	CNVR ID	Chr	Start	End	Size	Description	Adjusted *p*-Value
MY	CNVR_155	17	71,644,239	71,885,069	240,831	Loss	0.031
	CNVR_156	17	72,803,799	73,155,293	351,495	Loss	0.033
	CNVR_106	11	103,230,152	104,256,742	1,026,591	Mixed	0.040
MFR	CNVR_124	14	146,715	891,340	744,626	Mixed	0.000
	CNVR_140	15	69,494,372	70,392,722	898,351	Gain	0.011
	CNVR_184	21	69,165,607	69,778,711	613,105	Loss	0.012
	CNVR_32	3	63,079,015	65,566,990	2,487,976	Gain	0.015
MFY	CNVR_148	17	22,459,660	23,169,353	709,694	Gain	0.01
	CNVR_106	11	103,230,152	104,256,742	1,026,591	Mixed	0.018
	CNVR_113	12	58,857,555	60,002,395	1,144,841	Gain	0.021
	CNVR_124	14	146,715	891,340	744,626	Mixed	0.028
	CNVR_156	17	72,803,799	73,155,293	351,495	Loss	0.030
	CNVR_161	18	48,610,254	48,869,465	1,026,591	Gain	0.047
MPR	CNVR_106	11	103,230,152	104,256,742	1,026,591	Mixed	0.013
	CNVR_155	17	71,644,239	71,885,069	240,831	Loss	0.020
	CNVR_156	17	72,803,799	73,155,293	351,495	Loss	0.035
	CNVR_107	11	104,418,358	106,541,521	2,123,164	Mixed	0.037
	CNVR_59	5	119,074,417	119,998,002	923,586	Mixed	0.042
	CNVR_40	3	109,548,750	110,571,030	1,022,281	Mixed	0.048

CNVR ID = CNVR number; Chr = chromosome number; Start = CNVR start position; End = CNVR end position; Size = CNVR size; Adjusted *p* value = adjusted *p* value.

**Table 5 animals-14-00017-t005:** Details of the KEGG pathways significantly enriched from genes in CNVRs.

Pathway	Description	Gene Name	FDR(q)-*p*-Values
bta04920	Adipocytokine signaling pathway	AKT1, NFKBIB, RXRA, TRAF2, CPT1B	0.0003
bta01522	Endocrine resistance	NOTCH1, AKT1,JAG2, MAPK12, MAPK11	0.001
bta04658	Th1 and Th2 cell differentiation	NOTCH1, JAG2, NFKBIB, MAPK12, MAPK11	0.0013
bta04664	Fc epsilon RI signaling pathway	AKT1, VAV2, MAPK12, MAPK11	0.0026
bta04621	NOD-like receptor signaling pathway	CARD9, SHARPIN, NFKBIB, TRAF2, MAPK12, MAPK11	0.0051
bta00513	Various types of N-glycan biosynthesis	MAN1B1, MAN1B1, ALG12	0.0052
bta00510	N-Glycan biosynthesis	MAN1B1, MAN1B1, ALG12	0.0088
bta00564	Glycerophospholipid metabolism	AGPAT2, PLD4, CHKB	0.0104
bta04625	C-type lectin receptor signaling pathway	CARD9, AKT1, MAPK12, MAPK11	0.0107
bta04622	RIG-I-like receptor signaling pathway	NFKBIB, TRAF2, MAPK12, MAPK11	0.0111
bta04370	VEGF signaling pathway	AKT1, MAPK12, MAPK11	0.0119
bta04659	Th17 cell differentiation	NFKBIB, RXRA, MAPK12, MAPK11	0.015
bta04071	Sphingolipid signaling pathway	AKT1, TRAF2, MAPK12, MAPK11	0.0168
bta04722	Neurotrophin signaling pathway	AKT1, NFKBIB, MAPK12, MAPK11	0.0178
bta00601	Glycosphingolipid biosynthesis—lacto and neolacto series	ABO, FUT7	0.0249
bta04917	Prolactin signaling pathway	AKT1, MAPK12, MAPK11	0.0315
bta04015	Rap1 signaling pathway	AKT1, VAV2, GRIN1, MAPK12, MAPK11	0.0326
bta04914	Progesterone-mediated oocyte maturation	AKT1, MAPK12, MAPK11	0.0376
bta00250	Alanine, aspartate and glutamate metabolism	GPT, ADSS1	0.0387
bta00350	Tyrosine metabolism	COMT, DBH	0.0405

## Data Availability

Data are available upon request from the authors.

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
