# Peer review of "Descriptive Statistics and Genome-Wide Copy Number Analysis of Milk Production Traits of Jiangsu Chinese Holstein Cows"

_animals, 2023, doi:10.3390/ani14010017_

Round 1
Reviewer 1 Report
Comments and Suggestions for Authors
The document has many typos, especially in spaces/punctuation problems, for example, L224, 226 Spaces.
In materials and methods, authors do not give all information, for example, the number of records, is one per animal, per lactation or per trait? or they have adjusted values for lactation, or they used multiple records per animal per lactation. The number of records and animals does not fit. For GWAS, The phenotypes were lactations adjusted to 305 days or genetic breeding values, please specify and include the used model.
In order to understand the importance of CNV contribution, it should be important to include the percentage of genetic variance explained by CNV or determine the descripcion and its aportation.
In conclusion, I would like to see the inclusion of perspectives or implications of including this information for productive traits.
Author Response
Comment 1: The document has many typos, especially in spaces/punctuation problems, for example, L224, 226 Spaces.
In materials and methods, authors do not give all information, for example, the number of records, is one per animal, per lactation or per trait? or they have adjusted values for lactation, or they used multiple records per animal per lactation. The number of records and animals does not fit. For GWAS, Phenotypes were lactations adjusted to 305 days or genetic breeding values, please specify and include the used model.
RE: Thanks for the comments on this study. We have added relevant information in the Materials and Methods section. The phenotypes were lactations adjusted to 305 days, we have added previously published papers as references, and the calibration model used in the literature is the one used in this study.
Comment 2: In order to understand the importance of CNV contribution, it should be important to include the percentage of genetic variance explained by CNV or determine the descripcion and its aportation.
In conclusion, I would like to see the inclusion of perspectives or implications of including this information for productive traits.
RE: Thanks for the comments on this study. We have added relevant content in the discussion section.
Reviewer 2 Report
Comments and Suggestions for Authors
General comments
In this manuscript, the authors described the relationship between copy number variation and milk production traits. This is an interesting study which provides useful information to genetic selection for high milk production.
Special comments
1. The high milk production may affect the productive lifespan and the subsequent profit, which was not addressed in the present study. Suggest including this limitation in your discussion.
2. Line 18. Change “Were” to “were”.
3. Abstract. Number of the cows and sample types used for genomic study and methods for CNV and SNP detection should be include in the abstract.
4. Lines 81-83. Rephrase. Suggest changing to “... are caused by SNPs and have an important....”.
5. Lines 130-131. Change “The whole process of hair sample collection and phenotypic trait data collection was 130 measured according to….” to “The procedures of hair sample collection and phenotypic trait data collection were carried out according to…”.
6. Lines 132-133. Change “was also accepted” to “was approved”.
7. Line 141. Change "is” to “was”.
8. Line 167. Did you use DNA? If so, the methods of extraction and its quality should be given.
9. Lines 168-169. Supplier of the chip needs to be given.
10. Lines 182-209. Grammatic check. Past tense should be used to describe the methods. What does CNVR stand for? Full name should be given at the first mention.
11. Line 219. What type of FDR methods was used, BH or q-value?
12. Line 227. Delete “were retained”.
13. Table 2. What type of correlation parameter was used in this table, R or R-square? If you used R, some of the correlation may have a negative label.
14. Lines 263-264. The annotations of "triangles" can not be found in the table above.
15. Table 5. Could you indicate Raw P or FDR-P was in the table?
16. Figure 4. The annotations are difficult to read.
17. Lines 346-348. The negative correlations were not shown in your Results
Comments on the Quality of English LanguageBasically the article reads well. But a moderate edit of English language is required.
Author Response
Comment 1: In this manuscript, the authors described the relationship between copy number variation and milk production traits. This is an interesting study which provides useful information to genetic selection for high milk production.
RE: Thanks for the comments on this study.
Comment 2: The high milk production may affect the productive lifespan and the subsequent profit, which was not addressed in the present study. Suggest including this limitation in your discussion.
RE: Thanks. The aim of this study was to describe the basic status of milk production traits of Chinese Holstein cows in Jiangsu, and locate the QTLs and functional genes that affect the five milk production traits of first-born cows, the relationship between the productive life and subsequent profit of dairy cows and milk production was not considered in this study, So we have no analization in the discussion section
Comment 3: Line 18. Change “Were” to “were”.
RE: Thanks. We have change it in the revised manuscript.
Comment 4: Abstract. Number of the cows and sample types used for genomic study and methods for CNV and SNP detection should be included in the abstract.
RE: Thanks for the comments on this study. We have added the number of the cows and sample types used for genomic study and methods for CNV and SNP detection.
Comment 5: Lines 81-83. Rephrase. Suggest changing to “... are caused by SNPs and have an important....”.\
RE: Thanks. We have revised the sentence in the revised manuscript.
Comment 6: Lines 130-131. Change “The whole process of hair sample collection and phenotypic trait data collection was 130 measured according to….” to “The procedures of hair sample collection and phenotypic trait data collection were carried out according to…”.
RE: Thanks. We have revised the sentence in the revised manuscript.
Comment 7: Lines 132-133. Change “was also accepted” to “was approved”
RE: Thanks. We have revised it in the revised manuscript.
Comment 8: Line 141. Change "is” to “was”.
RE: Thanks. We have revised it in the revised manuscript.
Comment 9: Line 167. Did you use DNA? If so, the methods of extraction and its quality should be given.
RE: We used DNA. We send the sample to the company for testing instead of extracting DNA by ourselves. We have indicated in the revised manuscript.
Comment 10: Lines 168-169. Supplier of the chip needs to be given.
RE: We have added the supplier of the chip in the revised manuscript.
Comment 11: Lines 182-209. Grammatic check. Past tense should be used to describe the methods. What does CNVR stand for? Full name should be given at the first mention.
RE: Thanks. We have checked grammatic and revised these in the revised manuscript. CNVR means CNV Region,we have revised it.
Comment 12: Line 219. What type of FDR methods was used, BH or q-value?
RE: Thanks. We used q-value.
Comment 13: Line 227. Delete “were retained”.
RE: Thanks. We have revised it in the revised manuscript.
Comment 14: Table 2. What type of correlation parameter was used in this table, R or R-square? If you used R, some of the correlation may have a negative label.
RE: We used R-square in the Table 2.
Comment 15: Lines 263-264. The annotations of "triangles" can not be found in the table above.
RE: Thanks. We have revised annotations in the revised manuscript.
Comment 16: Table 5. Could you indicate Raw P or FDR-P was in the table?
RE: Thanks. We have indicated that the P value in the table is FDR-P.
Comment 17: Figure 4. The annotations are difficult to read.
RE: We have revised annotations in the revised manuscript.
Comment 18: Lines 346-348. The negative correlations were not shown in your Results
RE: Thanks. We have revised the paragraph in the revised manuscript.
Comment 19: Basically the article reads well. But a moderate edit of English language is required.
RE: Thanks. We have edited English language in the revised manuscript.
Round 2
Reviewer 1 Report
Comments and Suggestions for Authors
NA
Author Response
Thanks for your agreement with our revised version.
Reviewer 2 Report
Comments and Suggestions for Authors
In this revised version, the authors incorporated some of my comments. However the following comments need to be considered.
1. Line 131. Change “The procedures of hair sample collection and phenotypic trait data collections was” to “The procedures of hair sample and phenotypic trait data collection were”.
2. Line 169-170. Change “with not less than” to “with at least”.
3. Line 197. Change “use” to “used”.
4. Line 199. Change “are” to “were”
5. Lines 199, 200, 201. Change “should be” to “was”.
6. Line 204. Change “are” to “were”.
7. Line 206. Change “have” to “had”, “are” to "were”.
8. Line 207. Change “overlaps” to “overlapped.
9. Line 208. Change “are” to “were”.
10. Line 209. Change “take” to “took”.
11. Line 212. Change “will be used” to “were used”.
12. Line 223. Change “FDR” to “FDR (q-value)”.
13. Table 2. Please indicate you used R-square in table 2.
14. Table 2. There are not diagonal lines in table 2. Please add it.
15. Table 5. Change “Adjusted p values” to “FDR(q)-p values”.
16. Figure 4. The GO annotations are difficult to read. Please use larger fonts”.
Comments on the Quality of English LanguageA moderate edit of English language is required.
Author Response
Comment 1: Line 131. Change “The procedures of hair sample collection and phenotypic trait data collections was” to “The procedures of hair sample and phenotypic trait data collection were”.
RE: Thanks. We have changed the sentence in the revised manuscript.
Comment 2: Line 169-170. Change “with not less than” to “with at least”.
RE: Thanks. We have changed it in the revised manuscript
Comment 3: Line 197. Change “use” to “used”.
RE: Thanks. We have changed it in the revised manuscript.
Comment 4: Line 199. Change “are” to “were”
RE: Thanks. We have changed it in the revised manuscript.
Comment 5: Lines 199, 200, 201. Change “should be” to “was”.
RE: Thanks. We have changed these in the revised manuscript.
Comment 6: Line 204. Change “are” to “were”
RE: Thanks. We have changed it in the revised manuscript.
Comment 7: Line 206. Change “have” to “had”, “are” to "were”.
RE: Thanks. We have changed it in the revised manuscript.
Comment 8: Line 207. Change “overlaps” to “overlapped.
RE: Thanks. We have revised it in the revised manuscript.
Comment 9: Change “are” to “were”
RE: Thanks. We have revised it in the revised manuscript.
Comment 10: Line 209. Change “take” to “took”
RE: We have changed it in the revised manuscript.
Comment 11: Line 212. Change “will be used” to “were used
RE: Thanks. We have We have changed it in the revised manuscript.
Comment 12: Line 223. Change “FDR” to “FDR (q-value)”.
RE: Thanks. We have changed it in the revised manuscript.
Comment 13: Table 2. Please indicate you used R-square in table 2.
RE: Thanks. We have indicated it in the revised manuscript.
Comment 14: Table 2. There are not diagonal lines in table 2. Please add it.
RE: We have added diagonal lines in table 2 in the revised manuscript.
Comment 15: Table 5. Change “Adjusted p values” to “FDR(q)-p values”
RE: Thanks. We have revised it in the revised manuscript.
Comment 16: The GO annotations are difficult to read. Please use larger fonts”.
RE: Thanks. We have resupplied a large one.
Comment 17: A moderate edit of English language is required
RE: Thanks. We have edited the English language.